# Recent Advances in Bacteria-Based Cancer Treatment

**DOI:** 10.3390/cancers14194945

**Published:** 2022-10-09

**Authors:** Xianyuan Wei, Meng Du, Zhiyi Chen, Zhen Yuan

**Affiliations:** 1Faculty of Health Sciences, University of Macau, Taipa, Macau 999078, China; 2Centre for Cognitive and Brain Sciences, University of Macau, Taipa, Macau 999078, China; 3Institute of Medical Imaging, Hengyang Medical School, University of South China, Hengyang 421200, China

**Keywords:** tumor therapy, engineered bacteria, bacteria-based cancer treatment

## Abstract

**Simple Summary:**

Cancer refers to a disease involving abnormal cells that proliferate uncontrollably and can invade normal body tissue. It was estimated that at least 9 million patients are killed by cancer annually. Recent studies have demonstrated that bacteria play a significant role in cancer treatment and prevention. Owing to its unique mechanism of abundant pathogen-associated molecular patterns in antitumor immune responses and preferentially accumulating and proliferating within tumors, bacteria-based cancer immunotherapy has recently attracted wide attention. We aim to illustrate that naïve bacteria and their components can serve as robust theranostic agents for cancer eradication. In addition, we summarize the recent advances in efficient antitumor treatments by genetically engineering bacteria and bacteria-based nanoparticles. Further, possible future perspectives in bacteria-based cancer immunotherapy are also inspected.

**Abstract:**

Owing to its unique mechanism of abundant pathogen-associated molecular patterns in antitumor immune responses, bacteria-based cancer immunotherapy has recently attracted wide attention. Compared to traditional cancer treatments such as surgery, chemotherapy, radiotherapy, and phototherapy, bacteria-based cancer immunotherapy exhibits the versatile capabilities for suppressing cancer thanks to its preferentially accumulating and proliferating within tumors. In particular, bacteria have demonstrated their anticancer effect through the toxins, and other active components from the cell membrane, cell wall, and dormant spores. More importantly, the design of engineering bacteria with detoxification and specificity is essential for the efficacy of bacteria-based cancer therapeutics. Meanwhile, bacteria can deliver the cytokines, antibody, and other anticancer theranostic nanoparticles to tumor microenvironments by regulating the expression of the bacterial genes or chemical and physical loading. In this review, we illustrate that naïve bacteria and their components can serve as robust theranostic agents for cancer eradication. In addition, we summarize the recent advances in efficient antitumor treatments by genetically engineering bacteria and bacteria-based nanoparticles. Further, possible future perspectives in bacteria-based cancer immunotherapy are also inspected.

## 1. Introduction

Cancer refers to a disease involving abnormal cells that proliferate uncontrollably and can invade normal body tissue. According to the World Health Organization (WHO), cancer is the second leading cause of death worldwide [1]. It was estimated that at least 9 million patients are killed by cancer annually. Conventional therapies for cancer include surgery, chemotherapy, and radiotherapy. However, the downside of these traditional cancer treatment methods is that patients often suffer from various side effects during treatment. In particular, conventional treatment exhibits low specificity, leading to drug resistance in cancer cells.

Meanwhile, bacteria also have played an important role in maintaining good health and preventing diseases from healthier environments for millions of years. It is estimated that the human body contains trillions of bacteria [2]. The human gastrointestinal tract is the largest reservoir of commensal bacteria [3]. Intestinal bacteria such as Firmicutes, Bacteroides, Actinomycetes, and Enterobacteriaceae promote human health by synthesizing vitamin K, preventing colonization of pathogens, and maintaining the homeostasis of intestines [4]. There are more than 500 strains of bacteria such as *Streptococcus* and *Actinomycetes* in the mouth, which forms a protective biofilm on the surface of the teeth [5,6,7]. Additionally, Lactobacillus is dominant in the human vagina, maintaining the pH homeostasis of the environment by secreting lactic acid and inhibiting the interaction of other bacteria with epithelial cells [8]. By contrast, *Lactobacillus*, *Staphylococcus*, *Streptococcus*, etc., exist in the skin and nasal cavity, protecting the human body from other pathogens [7,9]. More importantly, recent studies have demonstrated that bacteria play a significant role in cancer treatment and prevention. In this review, we will firstly introduce the naïve bacteria and bacterial components of anticancer activity. We then summarize the recent work on efficient antitumor treatments that combine bacteria and nanoparticles. Further, we demonstrate that bacteria can be equipped with anticancer properties through gene editing technology, which provides a new insight into cancer therapy.

## 2. Bacterial Components of Antitumor Treatment

To date, bacterial toxins produced by bacterial cells, such as the Coley toxin, diphtheria toxin, Clostridium perfringens enterotoxin, bacterial enzymes L-asparaginase and arginine deaminase, and biosurfactant, such as surface and prodigiosin-like pixels, is able to effectively inhibit tumor growth through cell-cycle arrest, tumor-cell signal-pathway interruption, and other mechanisms. In addition, the components of bacteria, bacterial outer surface, the bacterial membrane, bacterial wall, and biofilm can also specifically activate the immune response to kill tumor cells.

### 2.1. Bacterial Toxins

In 1891, Dr. Coley successfully cured cancer patients with a mixture of live bacteria and “Coley toxin” heat-inactivated bacteria *Streptococcus pyogenes* and *Bacillus mirabilis*, opening the door to bacterial treatment of cancer [10,11]. Subsequent studies illustrated that Coley toxins include exotoxins produced by *Streptococcus pyogenes* and *Serratia marcescens*. In addition, *S. pyogenes* can produce pyrogenic exotoxins SpeA, SpeB, and SpeC, which have the ability to nonspecifically stimulate CD4+ lymphocytes, resulting in stronger secretion of different cytokines [12]. Similarly, prodigiosin, produced by *S. marcescens*, is a low molecular weight, red pigment, and heterocyclic tripyrrole toxin with antitumor activity, causing fever and potential antitumor immune response when combined with other components in the preparation [13].

Diphtheria toxin is a toxic protein produced by *Corynebacterium diphtheria*, while DTAT is its modified form, which targets the vascular endothelium of the tumor, results in the regression of cancer tissues in mice [14]. Clostridium difficile toxin includes two subtypes of cytotoxin (TcdB) and enterotoxin (TadA), which can kill cancer cells by recruiting proinflammatory factors to activate immune response [15]. Clostridium perfringens enterotoxin produced by *C. perfringens* also has anticancer activity, which leads to dose-dependent acute toxicity by binding to the overexpressed claudin-4 receptor on pancreatic cancer cells [16]. In addition, Verotoxin 1 (vt-1) is produced by pathogenic *Escherichia coli* and its function is to arrest the cancer cell cycle. Exotoxin A (PE) synthesized by *Pseudomonas aeruginosa* inhibits protein synthesis through ADP ribosylation, leading to cancer cell apoptosis [17,18]. The recombinant protein has better anticancer activity by modifying the cell structure recognition domain of the protein and preserving the membrane translocation domain and ADP ribosylation domain [15]. Hemolysin produced by bacteria, such as hemolysin A produced by *E. coli clyA* gene and hemolysin O produced by *Listeria monocytogenes*, is toxic to cancer cells. As a bacterial virulence factor, *Listeria monocytogenes* is released from phagocytes by perforating the phagocyte membrane. This phagosome escape mechanism enables *Listeria monocytogenes* to finally induce the immune response through MHC class I molecules in the cytoplasm with the protein expressed by the vector as an endogenous antigen [19].

### 2.2. Bacterial Enzymes

Bacterial enzyme L-asparaginase from *Escherichia coli* is an effective cancer therapeutic agent, which can inhibit the progression of malignant cells by activating asparagine hydrolysis and reducing its blood concentration, thereby causing toxicity to the MCF-7, HepG2, and SK-LU-1 cell lines. Bacterial-derived asparaginase has been approved for the treatment of acute lymphoblastic leukemia and non-Hodgkin’s lymphoma [20]. In addition, Fiedler et al. demonstrated that *Streptococcus pyogenes* produces arginine deaminase, which can consume arginine in tumor cells, resulting in decreased proliferation of arginine-deficient tumor glioblastoma multiforme [21].

### 2.3. Biosurfactant

Cyclic lipopeptide is an example of a biosurfactant with extensive antibacterial and antitumor activities that is produced by *Bacillus subtilis natto TK-1*. Xiaohong Cao et al. demonstrated that cyclic lipopeptide inhibited proliferation of human breast cancer MCF-7 cells by inducing apoptosis and increasing ion calcium concentration in the cytoplasm. Flow cytometric analysis revealed that cyclic lipopeptide caused dose- and time-dependent apoptosis through cell arrest at G(2)/M phase [22]. Another lipopeptide such as surfactin, have also been demonstrated their potential antitumor activity against several cancer cell lines. [23]. Surfactin induces the increase in calcium ions in human breast cancer MCF-7 cells and the accumulation of tumor suppressor p53 and cyclin kinase inhibitor p21, leading to cell-cycle arrest and apoptosis [22]. The same genus *Marine Bacillus subtilis* sp., can also produce an L-lysine biopolymer Epsilon-poly-L-lysine with antibacterial and anticancer activity. Studies have shown that Epsilon-poly-L-lysinet has obvious cytotoxicity on the cervical adenocarcinoma cell HeLaS3 and liver cancer cell HepG2 [24]. *Pseudomonas libanensis m9-3* produces a cyclic lipopeptide named viscosin with extensive antibacterial and antitumor activities. The MTT results indicated that viscosin inhibited the proliferation of MDA-MB-231 in breast cancer at 15 uM concentration. Moreover, viscosin also inhibited the migration of the prostate cancer cell line PC-3M [25,26]. The cyclic peptide AT514 (serratamolide) from *Serratia marcescens* is cytotoxic to B-cell chronic lymphocytic leukemia and induces endogenous apoptosis by activating the release of caspase-3, the antitumor function of which was confirmed in mouse experiments [27]. A variety of secondary metabolites, prodigiosin-like fragments, BE18591 and roseophilin with antitumor activity were isolated from *Streptomyces* sp. BE18591 inhibited the growth of the human Thomas cancer cell MKN-45 [28]. Roseophilin binds to the intracellular antiapoptotic receptor Mcl-1 and induces apoptosis of cancer cells [29,30]. Prodigiosin-like fragments showed significant cytotoxic activity against the colon cancer cell line HCT-116, liver cancer cell line HepG-2, and breast cancer cell line MCF-7. Table 1 lists biosurfactants with cancer cell proliferation, which are known as antitumor agents and inhibit some cancer progression processes. Biosurfactants show promising application in microemulsion-based drug formulations. Microemulsion comprises an aqueous phase, an oil phase, and a surfactant, which can encapsulate or solubilize a hydrophobic or hydrophilic drug for antitumor therapy. The combination of biosurfactant and liposome also demonstrates specific targeting to cancer cell. Shim, Ga Yong et al. revealed that surfactin enhanced cellular delivery of liposome siRNA in Hela cells. In this way, it was possible to improve the antitumor effectiveness of those nanoparticles [31]. Biosurfactants have application in broad-spectrum antitumor treatments and are viewed as safe vehicles or ingredients in drug-delivery systems.

### 2.4. Extracellular Surface

Exopolysaccharides (EPS) are carbohydrate compounds secreted by Gram-positive *Lactobacillus* bacteria outside the cell wall and usually infiltrated into the culture medium in the process of growth and metabolism. Some adhere to the microbial cell wall to form a capsule, which are called capsular polysaccharides. They have a dose-dependent and time-dependent antitumor effect of antiproliferation, and they promote apoptosis in anticancer activity [32]. The S-layer that is composed of protein and glycoprotein on the outermost cell surface of Gram-positive bacteria also has antitumor activity. Studies have shown that the S-layer protein of *Lactobacillus acidophilus* CICC 6074 can be regarded as a potential antitumor drug [33].

### 2.5. Bacterial Cell Membrane

The bacterial membrane components used in anticancer treatment include the cytoplasmic membrane vesicles of Gram-positive bacteria and the outer membrane vesicles of negative bacteria, as well as membrane fragments. Because of its rich pathogen-associated molecular patterns (PAMPs), the bacterial membrane is recognized by antigen-presenting cells (APCs) and activates the immune activity of T cells. It can also bind and activate the toll-like receptor (TLR), which regulates the production of proinflammatory cytokines, such as IL-12, and other constituent molecules, such as CD40. Subsequently, these mediators produce interferon (IFN)-γ and start the Th1-dependent immune response, mainly mediated by CD8+ effector cells, which induces a strong immune response against cancer cells in the tumor microenvironment [34,35].

The results of Min Li et al. showed that the PAMP on *E. coli* outer membrane vesicles (OMVs) was effectively recognized and internalized by neutrophils in revascularization. Neutrophils then crossed the blood vessels and guided OMVs to target inflammatory tumors (Figure 1) [36].

The OMVs of Gram-negative bacteria are mostly used in anticancer research and are composed of a lipopolysaccharide (LPS), outer membrane protein (OMP), and PG similar to the outer membrane. β-Barrel assembly machine (BamA) protein guides and inserts the outer membrane protein of the OMV into the outer membrane of newly produced PG to induce the outer membrane maturation of cells [37]. Knocking out the phospholipid transporter vacj/yrb increases the production of OMVs in two Gram-negative bacteria: *Haemophilus influenzae* and *Vibrio cholerae* [38]. RNA binding protein L7ae and lysosomal escape protein listeriolysin O are modified on the surface of bacterial OMVs. L7ae specifically binds to the mRNA vaccine to deliver antigens to dendritic cells (DCs). Listeriolysin O mediates the phagosome escape mechanism. This OMV-based mRNA tumor vaccine delivery platform can significantly inhibit the growth of melanoma and colon cancer cells [39]. OMVs of Gram-negative bacteria hybridize with tumor-derived cell membranes (MTs) to form new functional vesicles. In vivo experiments showed that MT-OMVs can induce adaptive immune response and then inhibit lung metastasis of the tumor [40].

As a tool for the delivery of nanomaterial and vaccines, the bacterial membrane plays not only the role of antitumor cells, but also a role in antivirus and antibacterial infection. Hydrophobic drugs can be loaded through the incubation method with convenient operation, and common drugs can be loaded through electroporation, ultrasonic method, extrusion method, freezing cycle, and saponin treatment method. The bacterial membrane can be loaded with anticancer compounds, functional small RNA molecules, cancer cell antibodies, and cytokines, and can jointly eliminate cancer cells through chemotherapy, gene silencing or mutation, immunity, photothermal therapy, photodynamic therapy, and other methods. Bacterial extracellular vesicles (BEVs) existing in the tumor microenvironment can be engulfed by cells through antigen–antibody-specific binding or membrane fusion. The BEV then releases the cargo in the cytoplasmic space, allowing its nanomaterials to play a role leading to apoptosis, necrosis, or autophagy of cancer cells. For example, RNA drugs and antisense oligonucleotides were loaded into extracellular vesicles by ultrasound, which played a role in the mouse breast cancer model [41].

Different pathogens will prefer colonization of specific tissues, such as *Klebsiella pneumonia* infecting the lungs, *Neisseria meningitidis* and *Listeria monocytogenes* infecting brain nerves, which means specific bacterial membranes can be used to target the corresponding cancer sites. After incubating the OMV of *Klebsiella pneumonia* with doxorubicin, a broad-spectrum chemotherapy drug, and then mixing in PBS and removing the free doxorubicin, it quickly reached the vicinity of the lung tumor in A549 BALB/c nude mice, and TUNEL results indicated that it significantly induced tumor cell apoptosis (Figure 2) [42].

### 2.6. Cell Wall

The bacterial cell wall is mainly composed of peptidoglycans. In addition to maintaining the shape of bacterial cells, peptide aggregation was also found to be related to regulating immune response, stimulating the production of tumor necrosis factor, interferon, and interleukin (IL-1, IL-6, IL-8, IL-12) [43].

### 2.7. Biofilm

Biofilm formed by bacteria also plays a role in tumor treatment. Biofilm is a glycoprotein lipid layer spontaneously formed by bacteria in the process of growth and attached to abiotic or biological surfaces including protein, DNA, metabolites, and so on. Biofilm forms in the tumor microenvironment and inhibits the growth, metastasis, and diffusion of tumor cells. The anticancer metabolites secreted by different bacteria are released in the biofilm, which can be accumulated and retained, so that they can be transferred to play a role in tumor cells [44,45].

### 2.8. Dormant Spores

In a harsh environment, bacteria will produce dormant spores. The spores of Clostridium can resist the harsh external environment and only revive when targeting the hypoxic tumor microenvironment. These advantages make anaerobic spores one way to target cancer cells. Studies have shown that *Clostridium novyi NT* spores do not contain lethal toxins, will not cause any systemic side effects in the injected host, and are an effective therapeutic agent for experimental tumors in mice [46]. The spores of *Clostridium* spp. are transformed into the active state in the tumor microenvironment only because of their strict anaerobic nature and are used in cancer treatment [47].

### 2.9. Magnetosomes

Magnetosomes are unique prokaryotic organelles containing 35 and 120 nm sized magnetite (Fe_3_O_4_) or cinerite (Fe_3_S_4_) magnetic iron mineral crystals surrounded by phospholipid bilayers. Mature magnetosomes are arranged in chains in the bacterial cytoplasm to form magnetosome chains, which cause magnetotactic bacteria to swim in the direction of the Earth’s magnetic field line [48,49]. Magnetotropic bacteria (MTB) are natural biomineralized bacteria that synthesize multiple magnetic nanoparticle chains in their own cytoplasm and can sense external magnetic fields. In order to detect and treat cancer, the bacteria can be combined with chemotherapy or radiotherapy drugs to target their delivery through magnetic force [50].

## 3. Naïve Living Bacteria for Anticancer

Compared with the extracted bacterial components, anaerobic or facultative anaerobic naïve living bacteria have better tumor targeting. When naïve living bacteria enter the host body, they can actively target the tumor microenvironment, which includes the characteristics of hypoxia, high purine, and low acid content. They can deliver the anticancer bacterial components more efficiently. The bacteria with anticancer activity that have been verified by animal experiments include probiotics: *Lactobacillus* [51,52,53,54,55,56,57,58] and *Escherichia coli* [59], *Streptococcus*, *Lactococcus lactis*, *Lactobacillus casei*, *Lactobacillus casei Zhang*, *Bifidobacterium longum*, and *Clostridium butyricum*, etc.; general toxic bacteria include *Mycobacterium tuberculosis* [60], *Salmonella typhimurium* [61], and *Listeria monocytogenes* [62]; and pathogenic bacteria such as *Vibrio parahaemolyticus*, *Pseudomonas aeruginosa* [63], etc.

### 3.1. Mycobacterium tuberculosis

The first bacterial agent approved by the FDA was *Bacillus Calmette*–*Guerin* (BCG). The Pasteur strain was obtained by Calmette and Guérin after 231 passages in cultures as a vaccine for the prevention of tuberculosis. After entering the host, APC macrophages selectively ingest *Mycobacterium tuberculosis* and activate the powerful ability of the innate T-like effector cell group CD4+/CD8+ T cell. Weakened *Mycobacterium tuberculosis* is used to treat superficial nonmuscle invasive bladder cancer (NMIBC), which eventually leads to tumor cell apoptosis by activating the toll-related caspase-8 signaling pathway-like receptor 7 (TLR7). So far, this vaccine is still the most effective treatment for the disease [60,64,65].

### 3.2. Listeria monocytogenes

As a Gram-positive bacterium, *Listeria monocytogenes* is a superior carrier for cancer cell antigen delivery. It is absorbed by macrophages in the process of infection and synthesizes Listeria hemolysin O as a bacterial virulence factor, which destroys the integrity of the phagosome together with bacterial phospholipase. Listeria monocytogenes are released from phagocytes. This phagosome escape mechanism uses the *Listeria monocytogenes* protein as an endogenous antigen to finally induce CD4+ and CD8+ T-cell immune responses against tumors through MHC class I molecules [62,66,67].

### 3.3. Salmonella typhimurium

It was first reported in 1935 that *Salmonella typhimurium* has a high-efficiency antitumor effect. With in-depth study, it was found to enhance innate and adaptive anticancer immune responses through an inflammatory response. The specific enrichment ability of *Salmonella typhimurium* in the tumor hypoxic microenvironment is 1000 times greater than in normal tissues [61]. After *Salmonella typhimurium* is phagocytized by immune cells in vivo, because it lacks a phagosome escape mechanism, its surface protein is presented by MHC class II molecules as an exogenous antigen to induce CD8+ T cells and CD4+ T-cell cancer immune response (Figure 3).

### 3.4. Lactic Acid Bacteria

Lactic acid bacteria are a kind of probiotic Gram-positive bacteria that include *Lactococcus* and *Lactobacillus*. Spherical lactic acid bacteria include *Streptococcus*, *Lactococcus lactis*, and *Pediococcus*. *Lactobacilli* include *Lactobacillus rhamnosus*, *Lactobacillus casei*, and *Bifidobacterium longum*. *Streptococcus* is a probiotic that kills tumor cells by activating host immunity and has been verified in animal experiments and clinical experiments [51]. *Lactococcus lactis* produces Nisin A, inhibits the growth of colon cancer tumors, and stops the cell division cycle [52]. *Lactobacillus casei* produces antimicrobial peptide kl15; interleukin (IL) -22 cytokines are downregulated, and caspase-7 and caspase-9 are upregulated, inducing apoptosis of colon cancer cells and the host immune response [52]. The oral probiotics *Lactobacillus casei Zhang* increase the levels of short-chain fatty acids and nicotinamide in the serum and kidney, which reduces the damage to kidney cells [53]. *Lactococcus lactis* and *Streptococcus bovis* produce Nisin A and bovicin HC5, respectively, which kill breast cancer cells. In addition, antimutagenic and anti-inflammatory effects from lactic acid bacteria were also found [54,55].

*Bifidobacterium longum* is a facultative anaerobic probiotic strain. After intravenous injection of mice, it was found to specifically locate solid tumors and slow tumor growth. Anticancer treatment seems to increase TNF-α Cytokine and nitric oxide synthesis [56,57,58].

Spores-dex is prepared by the chemical reaction between *Clostridium butyricum* and glucan. The spores-dex can specifically target colon cancer after oral administration. In the tumor microenvironment, *Clostridium butyricum* ferments glucan to produce anticancer short-chain fatty acids. In the subcutaneous tumor model of mice, the high-efficiency tumor inhibition ability of drug-loaded spores-dex was verified [68].

### 3.5. Escherichia coli

*Escherichia coli* is a model organism of Gram-negative bacteria. The bacteria mainly used for antitumor treatment are probiotic strains *Escherichia coli* Nissle 1917 (ECN) and wild-type MG1655, rather than serotype therapeutic strains. Genome analysis showed that ECN lacked virulence factors, such as α-. The expression of hemolysin, p-fimbriae adhesin, and adaptive factors do not produce any enterotoxins or cytotoxins related to pathogenic *Escherichia coli* strains, and it is an excellent targeted tumor vector [59].

### 3.6. Pseudomonas aeruginosa

*Pseudomonas aeruginosa* strain 1409 is a Gram-negative, pathogenic clinical strain that induces TC-1 cell necrosis by activating the TLR4 receptor, phosphorylation of RIP3, and activation of MLKl. Moreover, the necrotic tumor cells release HMGB1 to further induce the maturation and migration of DC cells. DC promotes the immune response of T cells by presenting tumor-related antigens, resulting in the large-scale death of tumor cells. *Pseudomonas aeruginosa* 1409 exerted a good therapeutic effect in the mouse TC-1 tumor model. However, it is worth noting that this bacterium is highly pathogenic, and it is resistant to three basic antibiotics. Thus, it is difficult to remove the colonization in the body, so it is not suitable for direct treatment of cancer [63].

## 4. Engineered Bacteria for Cancer Treatment

Engineered bacteria refer to expression exogenous protein in precise period and position. Despite the advantages of this, it is still limited by a few shortcomings that need to be urgently solved. The most important issue to be aware of is safety. Engineered bacterial therapeutics based on engineered bacteria refer to obtention of antibiotic resistance cassettes and ethical issues with transgenes, which presses challenges for the future of engineered bacteria therapy. Although direct injection of bacteria has an obvious killing effect on cancer cells, its side effects cannot be ignored. The research has shown that too high or too low a number of bacteria cause bacterial ecological imbalance [69], and the toxicity of bacteria is also harmful to normal tissues. The study by Cayetano Pleguezuelos-Manzano found that the occurrence of CRC in colorectal cancer was related to colistin produced by pathogenic *PKS + E. coli*. It is a toxin encoded and synthesized by a *PKS* island, which induces DNA double bond breakage and the death of host cells [70]. Bacteria are known as an autonomously disorganized and proliferating species. Its unique PAMPs and virulence factors induce an immune response in the human body, which promotes the killing of cancer cells by the human immune system. When the immune response is excessive, it threatens the patient’s life. In short, safe and reliable attenuated targeted bacteria, gene editing to express endogenous bacterial toxins, pigment proteases, etc., or exogenous drug precursor enzymes, antigen immune fragments, cytokines, anti-immune checkpoints, and noncoding RNA, among others, are used. It can push the bacterial treatment of cancer to a new level of low dose and high efficiency. The clinical trials for bacterial cancer treatments are summarized in Table 2. The *S. typhimurium* VNP20009 strain and *Clostridium novyi*-NT spores have entered a phase I clinical trial, which all revealed a promising antitumor effect. The famous *S. typhimurium* VNP20009 strain achieved the purpose of constructing attenuated and purine-deficient strains through *msbB* and *purL* deletion. After intravenously injecting 1 × 10^6^–1 × 10^9^ CFU/mL of *S. typhimurium* VNP20009 in 24 patients with metastatic melanoma, bacteria target purine-rich tumor regions and reduce the host’s nitric oxide and proinflammatory cytokines such as TNF-α and IL-1β. Unfortunately, no objective antitumor effect was observed. Engineering bacteria to specifically target tumors or the combinations of bacteria-based with antitumor protein will be applied in the future for therapeutic effect on tumors.

### 4.1. Engineered Bacteria to Achieve Detoxification

#### 4.1.1. Modify the Bacterial Outer Membrane

The surface of Gram-negative bacteria is wrapped by an outer membrane rich in a variety of PAMP pathogen recognition molecules, which is a source of bacterial virulence. Through recombination or covering the outer membrane structure, the bacterial immune response to the human body can be reduced, and the dose tolerated by the human body can be increased to fight cancer cells. Specifically, the structure of lipid A, the main component of the outer membrane, is underacetylated by knocking out the *msbB* gene [87]. The synthase of the LPS such as *rfaG* and *rfaD* [88] is knocked out, resulting in the production of a truncated LPS with incomplete structure and overexpression of MSHA flagellin and bacterial surface capsular polysaccharide CAP to reduce the exposure of outer membrane surface virulence factors.

In *Salmonella typhimurium* and *Escherichia coli*, gene editing to knockout the *msbB* gene is a common method of detoxification. *msbB* encodes the catalytic enzyme of the acylation process of lipid A, the main component of the outer membrane, and converts the acylation of the pentaacylated lipopolysaccharide into a complete hexaacylated lipopolysaccharide. The *msbB* gene was knocked out to produce underage-type pentaacylated lipopolysaccharide and endotoxic lipid A [89]. The famous *S. typhimurium* VNP20009 strain achieved the purpose of constructing attenuated and purine-deficient strains through *msbB* and *purL* deletion. Ultimately, it targets purine-rich tumor regions and reduces the host’s nitric oxide and proinflammatory cytokines such as TNF-α. This was verified in tumor experiments in mice and pigs and has been safely used in patients with metastatic melanoma and renal cancer in phase I clinical trials. However, no efficacy was observed [72].

To this day, PAMPs can also effectively reduce toxicity by covering the surface of bacteria. *Pseudomonas aeruginosa* mannose-sensitive hemagglutinin (PA-MSHA) is a Gram-negative bacterium that overexpresses MSHA flagella after gene editing. It weakens the toxicity of Pseudomonas aeruginosa by minimizing the exposure of surface virulence factors. Engineered PA-MSHA can inactivate the EGFR epidermal growth factor pathway signal in cancer cells and induce apoptosis. In the mouse model of bladder cancer, an injection of PA-MSHA effectively inhibited the growth of the tumor [90].

Tetsuhiro Harimoto et al. established a programmable cap expression system for bacterial surface capsular polysaccharides. The system regulates the extracellular biopolymer through the external inducer IPTG, so that the extracellular membrane is wrapped, temporarily avoiding the immune attack of the host and prolonging the circulation time of bacteria in the body. Their conclusion was proven in a mouse tumor model. By overexpressing ICAP, they were able to increase the maximum tolerated dose of bacteria by 10 times. They encapsulated *E. coli* strains, enabling them to escape the immune system and reach tumors. Because they did not administer IPTG in vivo, *E. coli* ICAP lost its encapsulation through time and was easier to eliminate from other parts of the body, thereby minimizing toxicity [91].

#### 4.1.2. Nutritional Deficiencies

Transforming bacteria into specific nutrition-dependent mutations can also effectively reduce toxicity and improve their antitumor activity. *Salmonella typhimurium* produces leucine and arginine synthesis-deficient trophic strain a1-r by knocking out *leu* and *arg* genes [92]. Attenuated and purine-deficient strain VNP2009 was constructed by *msbB* and *purL* deletion, targeting purine-rich tumor regions [93].

*AroA* gene mutation leads to the nutritional deficiency of aromatic amino acids in bacteria, which is considered safe and used widely in attenuated strains. The absence of *aroA* in *Pseudomonas aeruginosa* resulted in a tenfold increase in the safe dose of the bacterium compared with the wild-type [94]. Genetic engineering of *aroA* and *aroD* double mutant *Salmonella typhimurium* as reported by Yoon W.S. was used to treat mouse melanoma, resulting in 50% tumor regression [95]. M. Gabriela Kramer et al. constructed *Salmonella typhimurium* LVR01 of attenuated mice knockout *aroC* and virus vector particles expressing the IL-12 (sfv-IL-12) gene. When inoculated into the mouse model of advanced breast cancer metastasis, sfv-IL-12 showed an effective antiangiogenesis effect, and the combined effect of sfv-IL-12 and lvr01 could inhibit tumor growth and metastasis, finally prolonging the survival time. It is an effective antimetastasis therapy [96].

#### 4.1.3. Reduce Toxin Expression

Bacterial toxins as the main virulence factors downregulate or knockout the expression and synthesis of related toxin genes, which can greatly reduce the virulence of bacteria and improve the dose tolerance of the human body.

So far, gene-editing knockout of the Ppgpp gene in *Salmonella typhimurium* led to a defect in the synthesis of guanosine 5′-diphosphate-3′-diphosphate. This signal molecule participates in the expression of bacterial toxin genes and changes the structure of lipid A through the deletion of *relA* and *spoT*, which can achieve a 10^5^-fold detoxification effect [97]. After the lethal toxin genes *toxA* and *toxB* were knocked out by heat shock in *Clostridium novyii*, the bacterium turned into a nontoxic and safe strain. After intravenous injection of the active bacterium, it targeted the tumor area and attacked the cancer cells through cytokine aggregation immune cells, which eventually led to the reduction in tumor growth and its disappearance in mice [98]. The toxicity to normal cells was reduced by knocking out the exotoxin genes *exoS*, *exoT*, *aroA*, and *lasI* of *Pseudomonas aeruginosa* [99].

### 4.2. Engineered Bacteria to Targeting

So far, increasing tumor-targeting by genetic engineering can reduce the injection concentration of antitumor bacteria and improve their safety and antitumor efficacy. According to the characteristics of hypoxia, high purine, and weak acidity in the tumor microenvironment, endogenous inducible promoters with specific responses can be designed. Moreover, the addition of the exogenous inducers arabinose, salicylic acid, Tet, IPTG, ultraviolet light, light, heat, etc., at specific time and space can accurately induce the expression of substances with anticancer activity. Drugs can also be delivered to specific cancer cells by targeted cancer cell aptamers or proteins, such as the RGD membrane-penetrating peptide, nuclear localization signal, secretory system signal peptide, antibody, etc.

#### 4.2.1. Endogenous Inducible Promoter

It is well-known that inducible promoters exert precise spatiotemporal regulation of protein expression. Promoters responding to hypoxia, acidity, high purine, and bacterial density of the tumor microenvironment were designed based on their characteristics. This ensures that after the bacteria are enriched into the tumor microenvironment, they will express proteins with anticancer activity to reduce the toxicity to normal tissues.

Mitra Samadi et al. designed a hypoxia-inducible expression system using the *nirB* promoter of the hypoxia response of *E. coli* BW25133 to express anticancer protein, cardiac peptides, and GFP signal proteins. In in vivo experiments, this inhibited the growth and metastasis of mouse breast cancer tumors and improved the survival rate of the mice [100]. In addition, the hypoxia-responsive *fdhF* promoter in *E. coli* mc1061 can be used to accurately regulate the expression of anticancer compounds [101].

The weak acidity of the tumor microenvironment enables acid response promoters to express active anticancer proteins. Kelly Flentie screened five genes—*adiY*, *yohJ,* STM1787, TM1791, and STM1793—related to the acidic environment of tumors by co-culturing the library of *Salmonella typhimurium* transposon insertion mutants with melanoma or colon cancer cells. The corresponding promoter, as an acidic promoter, seems to play a role in targeted tumors [102].

The construction of the nutrient-deficient strains mentioned in Section 4.1.2 enable acteria to specifically enrich within purine and amino acid tumor microenvironments. Purine-deficient strains were constructed by purl deletion to target purine-rich tumor regions [93]. *AroA* gene mutations can lead to nutritional deficiencies of aromatic amino acids in bacteria, targeting amino acid-rich tumor regions [95].

By controlling the expression of lysed proteins through the Luxi and LuxR quorum-sensing response system, the engineering flora can be lysed in microcapsules, and the expressed protein product can be released to the outside of the cell with the ova subunit vaccine using the bacterial microcomponent BMC as the carrier. A mouse subcutaneous tumor model was introduced to demonstrate the potential of nanoparticle delivery, successfully activate immunity in mice, and play a preventive role against b16-ova tumors [103]. The deletion of the quorum-sensing gene can reduce the toxicity of bacteria and increase their number at a mild dose. The *lasI* gene encodes the syntheses of the quorum-sensing homoserine lactones 3-oxo-c12-hsl and *rhlI* encodes C4-HSL occur in *Pseudomonas aeroginosa*. By deleting the *lasI* and *rhlI* genes, the number of bacteria in mild doses was increased 10 times compared with the wild-type [94].

#### 4.2.2. Exogenous Inducible Promoter

L-arabinose induced the araC promoter, salicylic acid induced the PM promoter, tetracycline induced the TET promoter; T7 promoter was induced by Isopropyl-β-D-thiogalactoside, single-stranded DNA repair protein RecA promoter was induced by ultraviolet light, and so on [104]. Protein expression promoters commonly used in bacteria have the ability to accurately induce engineering bacteria to express active antitumor proteins by in situ injection into the tumor microenvironment as exogenous inducers [105].

The activity of heat-sensitive promoter HSB was activated by ultrasound or light source stimulation. In wild-type *Escherichia coli* Nissle 1917, the temperature-sensitive promoter (HSB-GFP) plasmid was constructed to express the tumor necrosis factor α (TNF-α). The growth of 4T1 tumor mice injected with the bacterium was significantly inhibited after three heat-stimulation treatments [106].

Chunli Han designed a photogenetic scavenging gene circuit based on blue light-responsive OptO proteins EL222. The circuit adopted the strategy of dark inducing the expression of detoxifying protein CcdA and blue light inducing the expression of toxin protein CcdB. When exposed to a 488 nm laser, the engineering bacteria died. A photosensitive promoter is used to ensure that the engineering bacteria can be removed after use and to further ensure the safety of subcutaneous administration of engineering bacteria microcapsules. The survival of engineering bacteria microcapsules carrying photogenetic scavenging gene circuits after exposure to blue light in vitro or in vivo can be reduced by about three orders of magnitude [103].

#### 4.2.3. Signal Peptide

The reported signal peptides that improve the targeted effect of cancer cells include membrane-penetrating peptides, nuclear localization sequences, bacterial secretion system signal peptides, and so on.

By displaying RGD (Arg-Gly-ASP)-penetrating peptide on the surface of bacteria, the tumor was targeted and the therapeutic effects of attenuated Salmonella were enhanced [107]. Sujie Huang et al. linked the DNA toxin drug camptothecin (CPT) with the nuclear localization sequence to construct a nanomaterial with cancer nuclear localization ability. Experiments proved the enhancement of cytotoxicity and selectivity [108].

The special protein secretion system of bacteria delivers effector proteins to target eukaryotic cells through complex needle-like molecular machines. Type III and VI secretion systems have been used to secrete fusion proteins with anticancer activity owing to their widespread presence in bacteria [109,110]. In *Pseudomonas aeruginosa*, 54 amino acid fragments at the N-terminal of the ExoS protein for the Type III secretion system-mediated translocation were fused with fragments at the C-terminal of ovalbumin for antigen–antibody specific binding by genetic engineering. Then, based on the special Type III secretion system of *Pseudomonas aeruginosa*, it was injected into the host cells to induce the immune response of CD8+ T lymphocytes [94].

#### 4.2.4. Targeted Proteins

Current means to improve targeting cancer cells include engineering the expression of proteins with a high affinity with cancer cells. These toxin proteins or antibodies can specifically recognize cancer cell antigens 

OpcA protein can cross the blood–brain barrier and target nerve cells. Engineering expression of the outer membrane invasion protein OpcA of *Neisseria meningitidis* can guide the specific enrichment of *Neisseria meningitidis* in the central nervous system. Methotrexate, a chemotherapy drug, was loaded into manganese dioxide (MnO2) hollow nanoparticles with surface-modified *Neisseria meningitidis* OpcA protein to construct a bionic nanotreatment system with great potential for glioblastoma (MTX@MnO2-Opca) [111].

Through genetic engineering methods, 30 amino acids that specifically express the C-terminal of *Clostridium perfringens* enterotoxin (CPE) were constructed to specifically bind to cldn-4 antigen on the surface of cancer cells and induce apoptosis of cldn-4-positive cancer cells [112].

Engineering expression of cytolysin A (ClyA), located on the outer membrane surface, express with antibody fragmentof cancer cells can also improve targeting [113]. CD20 is a specific antigen that is overexpressed by lymphoma cells. The engineered single domain antibody expressing CD20 in Salmonella can significantly improve the tumor specificity of Salmonella [114].

#### 4.2.5. Aptamer

Aptamer is a single-stranded oligonucleotide synthesized by artificial screening, which specifically targets substrates such as small molecules, peptides, proteins, cells, and tissues. Zhongmin Geng et al. anchored the aptamer AS1411 on the surface of attenuated *Salmonella typhimurium* VNP2009 by amidation, which can specifically target the nucleolin nucleus overexpressed on the cancer cell membrane (Figure 4). In the tumor-bearing mouse model inoculated subcutaneously with 4T1 cancer cells, the accumulation of bacteria in the tumor tissue after 12 h was nearly two times higher than that of the nonanchored aptamers. In addition, the aptamer TLs11a, which has a high binding affinity with the hepatocellular carcinoma cell line, was also tested. After injection into mice, aptamer bacteria showed high enrichment of H22 cells and better inhibition of tumor growth [115].

### 4.3. Engineered Bacteria to Anticancer

#### 4.3.1. Drug Precursors and Drug Synthase

The gene of respiratory chain enzyme II NDH-2 was overexpressed in engineered *Escherichia coli* MG1655 to obtain a large amount of H_2_O_2_ (Figure 5.). The bacterial surface was covalently connected with magnetic Fe_3_O_4_ nanoparticles, which were then injected into animals to specifically colonize the tumor area and convert H_2_O_2_ into toxic hydroxyl radicals (OH) through the Fenton reaction, resulting in the increase in ROS and the induction of severe tumor cell apoptosis [116].

Cytosine deaminase has been successfully applied in two kinds of targeted pre-enzyme drug therapies in *Clostridium*. The first was gene editing *Clostridium* to overexpress cytosine deaminase and specifically convert the prodrug 5-fluorocytosine into the anticancer agent 5-fluorouracil at the tumor site. This anticancer effect was verified in tumor-bearing mice. The other was to overexpress the nitroreductase enzyme in Clostridium by gene editing. In the tumor microenvironment, CB1954 was converted into a DNA crosslinker with antibacterial and cancerous properties [117].

#### 4.3.2. Antibodies

The expression of immune checkpoint PD-1/PD-L1 and CTLA-4 inhibitors can block checkpoints and induce immune cells to activate a strong immune response. In addition, highly specific antibodies expressing tumor antigens can inhibit the growth of cancer cells. After gene editing, *C. novyi NT* and *C. sporogenes* are used to express the heavy chain variable region of the highly specific antibody of tumor antigen. This protein binds and inhibits the activity of HIF cells, reduces the expression of transcription factor hypoxia-inducible factor 1alpha, leads to the transformation of the tumor microenvironment from hypoxia to oxygen enrichment, slows the growth of cancer cells, and reduces the risk of cancer cell metastasis [118].

The gene expresses tumor-specific antigen NY-ESO-1 in *Salmonella typhimurium*, secretes it through the Type III protein secretion system, and presents the antigen to CD8+ T cells and CD4+ T cells through the MHC class I pathway, specifically activating the immune pathway against NY-ESO-1-positive cancer cells [119]. *Salmonella typhimurium* is presented by MHC class II molecules as an exogenous antigen to induce the immune response of CD8+ T cells and CD4+ T cells [119]. *Listeria monocytogenes* has the phagosome escape mechanism mentioned above. As an endogenous antigen, it finally induces the immune response through MHC I class molecules. Both bacteria have been widely studied as vaccine carriers in popular cancer treatment [120].

#### 4.3.3. Cytokines

Cell necrosis factor TNF-α is a double-edged sword, which can activate transcription factor nuclear factor NF-κb at a low dose. It can stimulate the proliferation of tumor cells, but it can also be used as a tumor suppressant at high doses. The gene-edited *Clostridium acetobutyricum* DSM792 expresses and secretes mouse TNF-α. The purpose of this experiment was to specifically target the tumor microenvironment and controllably regulate TNF- α secretion. Unfortunately, owing to the low level of bacterial colonization in the tumor microenvironment and the specific expression of TNF-α, a limited level failed to achieve the effect of tumor treatment [121]. The role of cytokine interleukin-2 (IL-2) is to kill tumor cells by activating natural killer cells and enhancing MHC-restricted T cells. However, high doses of IL-2 are toxic to normal tissues, so bacteria with active targeting are selected as carriers. The genetically edited *Clostridium acetobutyricum* DSM792 successfully slowed the growth of mouse tumors in animal experiments by specifically expressing mouse IL-2 [122]. Fas, a proapoptotic factor, can initiate apoptotic signals in cells and induce apoptosis of Fas-sensitive cells. Fas ligand FasL membrane protein was expressed in *Salmonella typhimurium* and injected intravenously into mouse d2f2 breast cancer or CT-26 colon cancer tumors. It was observed that the growth of primary tumors in mice was inhibited by 59% and 82%, respectively [123].

Hypoxia-inducible factor 1alpha is a transcription factor of genes responsible for cell survival triggered by hypoxia in the tumor microenvironment. Arjan J. Groot et al. inhibited tumor growth by expressing scFv of HIF-1α in *Clostridium* [118].

#### 4.3.4. Noncoding RNA

siRNA that interferes with the expression of target genes or miRNA that affects the post-transcriptional function of genes is loaded into bacterial microcapsules using electrotransfer, chemotransfer, etc. It improves the specificity of noncoding RNA and protects RNA from degradation during delivery. So far, CRISPR cas9 protein and sgRNA have only been delivered in cell vesicles and liposomes. For example, delivery through bacterial vesicles can more conveniently engineer the membrane surface to improve targeting and protection during delivery [124,125].

At present, Mir-16 mimic coated by nonliving bacterial minicells has inhibited tumor growth in animal models by restoring miRNA levels in tumor cells in phase 1 clinical trials [126,127]. Kinesin spindle protein (KSP) is overexpressed in tumor tissues and is an ideal candidate for targeted cancer therapy. Vipul Gujrati et al. loaded small interfering RNA targeting KSP into attenuated OMV by electroporation. After injecting OMV-packaged siRNA into tumor mice, obviously targeted gene silencing and tumor inhibition were indicated [128].

#### 4.3.5. Pigment Synthase and Fluorescent Protein

Vipul Gujrati et al. expressed rhizobium tyrosinase MelA protein in attenuated *E. coli*, which can metabolize tyrosine into natural melanin and accumulate, with bacteria obtaining an unexpected photothermal effect. It can monitor the distribution of bacteria in vivo under near-infrared light irradiation and photothermal treatment of 4T1 tumor-bearing mice [129]. Zhijuan Yang et al. expressed firefly luciferase in *Salmonella typhimurium* ΔppGpp (STΔppGpp) (Figure 6.). The bacterium and photosensitizer Ce6 were injected into large tumor rabbits, and D-fluorescein continuously produced light to stimulate Ce6 to produce exogenous ROS. Compared with the external 660 nm light excitation of Ce6 by traditional photodynamic therapy, the internal light source of D-fluorescein had better photodynamic effect to excite Ce6 and inhibit tumor growth [130].

Engineered bacteria express luciferase fused with the human influenza hemagglutinin tag (Luc–HA) and bioluminescence detection technology is used to visualize the distribution of bacteria in organisms [131] to achieve Lux fluorescein and fluorescence tracing. In addition, *E. coli* was injected into mice with tumor colonization. When facultative anaerobic *E. coli* actively targeted the tumor microenvironment, 18F fluorodeoxysorbitol (FDS) positron emission tomography (PET), which can be used for specific binding with *E. coli*, was injected to image the distribution of *E. coli* in vivo. This method is expected to be used for semiquantitative visualization of tumor-targeted bacteria [132]. Engineering bacteria expressing GFP, YFP, mCherry, and other fluorescent proteins are injected into the body through microcapsules for in vivo imaging observation. The results indicate that the fluorescence can be maintained in the body for no less than 15 days [103].

#### 4.3.6. Bacterial Toxins

Among the bacterial component anticancer 1.1 bacterial toxins, a variety that have been experimentally proven to have anticancer activity are listed, including the Coley toxin, diphtheria toxin, *Clostridium perfringens* enterotoxin, azurol, cyclic dipeptide, rhamnolipid, and cytochrome A.

Overexpression of high concentrations of cytolysin A (ClyA) in engineered bacteria can bind and form pores in eukaryotic cell membranes, triggering caspase-mediated programmed cell death [113]. Pei Pan et al. designed the engineering bacterium BAC to increase the tumor-targeted ability and overexpressed the cytolysin A (ClyA) protein to regulate the cell cycle from the antiradiation phase to the radiation-sensitive phase. It inhibited the growth of mouse breast cancer and reduced the side effects of radiotherapy [133].

Yale Yue et al. constructed a protein expression plasmid to fuse the tumor antigen (Ag) and Fc fragments of mouse immunoglobulin G (IgG) to the C-terminal of OMV (ClyA–Ag–MFC) surface protein ClyA. In situ controllable production of the OMV carrying the tumor antigen (OMV–Ag–MFC) was achieved in the intestine by oral administration of modified bacterial *E. coli* and expression-inducer L-arabinose. These OMV–Ag–MFCs effectively cross the intestinal epithelial barrier and are absorbed by DCS in the lamina propria, followed by lymph node drainage and tumor antigen presentation. In a variety of mouse cancer models, tumor antigen-specific immune activation significantly inhibited tumor growth and resisted tumor challenges [131].

## 5. Bacteria-Based Nanoparticles for Cancer Treatment

Bacteria and nanomaterials are directly antitumor through the covalent connection of chemical amide bonds and can also be adsorbed together through electrostatic interaction. In addition, bacterial membrane fragments and outer membrane vesicles can be fused with bacteria through repeated freezing and thawing, ultrasound, extrusion, and other methods. The new combination of nanomaterials and bacteria can inhibit the growth of tumors through the anticancer activities of nanomaterials such as photothermal and photodynamic therapy, which not only reduce the toxicity to normal cells, but also increase tumor specificity.

### 5.1. Chemical Bond Connection

So far, the way to a stable combination of bacteria and nanomaterials has been mainly through peptidoglycans of the bacterial cell wall or cell membrane liposomes to form stable amide bonds with nanomaterials. Through an amide condensation reaction, MnO_2_ was modified on the surface of *Shewanella oneidensis* MR-1 bacterial cells, which can specifically decompose lactic acid. *Shewanella oneidensis* MR-1 (*S. oneidensis* MR-1) continuously decomposes lactic acid produced by the glucose metabolism in the tumor microenvironment by transferring electrons to the metallic mineral MnO_2_ in an anaerobic environment. This inhibited the growth of CT26 tumor cells in mouse experiments [134].

Wencheng Wu et al. immobilized liposomes co-loaded with lactic acid oxidase (LOD) and prodrug tirapazamine (TPZ) on the surface of *lactic acid bacteria* (LA) through an amide condensation reaction (Figure 7). Tumor specificity by LA can effectively deliver drug substances to tumor tissues. Loaded lactic acid oxidase (LOD) catalyzes the oxidation of lactic acid to H_2_O_2_, increasing the level of oxidative stress, which further aggravates hypoxia in tumors, thereby activating the TPZ prodrug sensitive to hypoxia and inducing significant tumor cell apoptosis and immunogenic cell death ICD [135].

### 5.2. Electrostatic Interaction

The cell walls of Gram-positive bacteria are crosslinked by negatively charged N-acetylglucosamine and N-acetyl cell wall acid. LPS, the outer membrane component of Gram-negative bacteria, is usually negatively charged. Therefore, stable binding bacterial nanomaterials can be obtained by incubating the positively charged nanomaterials with negatively charged bacteria.

Di Wei Zheng et al. assembled carbon-dot-doped carbon nitride (C_3_N_4_) on the surface of engineered *E. coli* carrying a nitric oxide (NO)-generating enzyme by electrostatic interaction. Under the light, the photoelectrons produced by C_3_N_4_ can be transferred to *E. coli* and promote endogenous NO_3_^-^ metabolism to tumor cells. This method has achieved good therapeutic effects in mouse tumor models [136]. Shuaijie Ding connected black phosphorus quantum dots (BPQDs) to the surface of *E. coli* genetically engineered to express catalase by electrostatic adsorption, thereby producing an engineered *E. coli*/BPQD (EB) system (Figure 8). After intravenous injection into mice, EB can target hypoxic tumor tissues and produce reactive oxygen species to destroy bacterial membranes. The released catalase degrades hydrogen peroxide to produce oxygen to alleviate hypoxia in tumors, thereby enhancing BPQD-mediated photodynamic therapy. The system can effectively kill tumor cells in vivo [137].

### 5.3. Bacterial Loading Nanomaterials

So far, nanomaterials with antitumor activity have been combined with bacteria using liquid nitrogen through repeated heating, freezing and thawing, ultrasonic vibration, physical extrusion, buffer solution incubation, electroconversion, chemical conversion to form perforation, etc. The common feature is that nanomaterials are wrapped or wrapped to form biologically active nanomaterials. Yao Liu et al. constructed an immunotherapy system of a natural red blood cell (RBC) membrane wrapping *Listeria monocytogenes*, with virulence factors removed by the extrusion method (LMO@RBC) (Figure 9). The nanomaterial produces a low systemic inflammatory response, and its accumulation effect in tumors is also improved owing to the long-term blood circulation ability of RBCs and the tumor hypoxic microenvironment colonization ability of LMO. In the BALB/c solid tumor model, LMO@RBC reached the tumor microenvironment, promoted the release of ROS in the tumor area and the activation of caspase 8, inducing extensive porogen gasdermin C (GSDMC) and dependent pyroptosis, which showed a high inhibitory effect on the growth of primary tumors and distant tumors [138].

In addition, Jiayu Zhang et al. modified the PD-L1 antibody on the surface of a *Salmonella typhimurium* OMV by co-extrusion of a 200 nm polycarbonate membrane and added hydrophilic catalase CAT and hydrophobic photosensitizer Ce6 to the OMV. By alleviating tumor hypoxia and improving the photodynamic and immunotherapeutic effects, the tumor was significantly inhibited [139].

## 6. Conclusions

Bacteria function as mutually beneficial symbiotic partners with the human body. Great progress has been made in nearly a century since bacteria were first used to treat cancer in 1891, and some results have entered the stage of clinical evaluation.

In this review, we discussed the anticancer activity of bacterial cellular components, including bacterial specific virulence factors, bacterial toxins, bacterial enzymes, biosurfactant, the outer surface of Gram-positive bacteria, bacterial cell membranes, cell walls, biofilms formed by spontaneous aggregation, dormant spores formed by poor nutrition, and magnetosomes, among others. In addition, we discussed the direct anticancer effect of naïve living bacteria. For example, probiotic lactic acid bacteria and *Escherichia coli* are used to treat colon cancer, breast cancer, etc. The general pathogenic bacteria *Mycobacterium tuberculosis*, *Listeria monocytogenes*, and *Salmonella typhimurium* are used to treat nonmuscle invasive bladder cancer and breast cancer. The pathogen *Pseudomonas aeruginosa* inhibits tumor growth in the mouse TC-1 tumor model. However, the natural virulence factors in bacteria can activate relevant PAMP immune responses, and they are difficult to remove after colonization. Therefore, engineered bacteria after gene editing are more suitable for cancer treatment. Genetic engineering transformation methods are as follows: (1) Engineering bacteria that edit genes such as *msbB*, *rfaG*, and *rfaD* can modify the structure of the outer membrane of Gram-negative bacteria and reduce the toxicity to the host. (2) Through overexpression of MSHA flagella or ICAP capsular polysaccharide, the bacterial surface virulence factors are masked. (3) Transforming bacteria into specific nutrition-dependent mutations can also effectively reduce toxicity, such as *aroA* gene mutations that lead to bacterial aromatic amino acid nutritional deficiencies. (4) Downregulation or knockout of the expression and synthesis genes of bacterial toxin-related toxins, which are the main virulence factors, can greatly reduce the virulence of bacteria. (5) The expression of an inducible promoter has precise space–time regulation ability, which can improve the specificity of bacteria. Exogenous inducible promoters induce the expression of anticancer active ingredients such as environmental factors and endogenous inducible promoters. (6) Other ways to improve specificity are the expression of the membrane-penetrating peptide RGD by engineered bacteria and the secretion of the systemic effector protein signal peptide by bacteria. (7) Engineering bacteria-expressed antibodies with high affinity to cancer cell surface antigens or that connect with single-stranded oligonucleotide aptamers is one way to improve specificity. (8) Engineered bacteria can also treat tumors by expressing active anticancer ingredients, such as the expression of drug synthase, cancer cell antibodies, the immune checkpoint inhibition antibody anti-PD-L1, cytokines, siRNA, miRNA, Crispr-Cas9, pigment synthase, fluorescent protein, bacterial toxin, and so on. Bacteria and nanomaterials are combined through chemical bonds, amide bonds, electrostatic interaction, or extrusion ultrasound. This anticancer approach combines the active targeting by bacteria and the high-efficiency cancer cell lethality of nanomaterials, from the cellular components of bacteria to naïve living bacteria, and the combination of engineered bacteria and nanomaterials. The application of bacteria in cancer treatment has changed from weak specific and weak immune response to efficient specific and immune activation response, with significant anticancer activity.

Despite the advantages of bacteria-based antitumor therapy, it is still limited by a few shortcomings that need to be urgently resolved. The most important is effectiveness. Bacteria in anticancer treatment refer to precise targeting processes and complex human immune responses in tumor environments. Engineered bacteria need to cross complex blood vessels in high-speed blood flow to reach the tumor environment after intravenous injection. How to accumulate enough engineered bacteria to exert anticancer effects is an urgent consideration. Ensuring the highly efficient expression of proteins that exert anticancer effects for a long time has an important impact on bacteria-based anticancer therapy. Another issue to be aware of is safety. Bacteria are known as an autonomously disorganized and proliferating species. Their unique PAMPs and virulence factors induce an immune response in the human body, which promotes the killing of cancer cells by the human immune system. When the immune response is excessive, it threatens the patient’s life. Genetic material based on engineered bacteria also refers to ethical issues with transgenes, which presses challenges for the future of engineered bacteria therapy. In general, future development directions of bacterial tumor therapy will be to combine nanomaterials and engineering bacteria and to explore the physical and functional relationship between bacteria and nanomaterials. The results will be a more specific, effective, and accurate tumor immune response together with a comprehensive and less toxic treatment system.

## Figures and Tables

**Figure 1 cancers-14-04945-f001:**
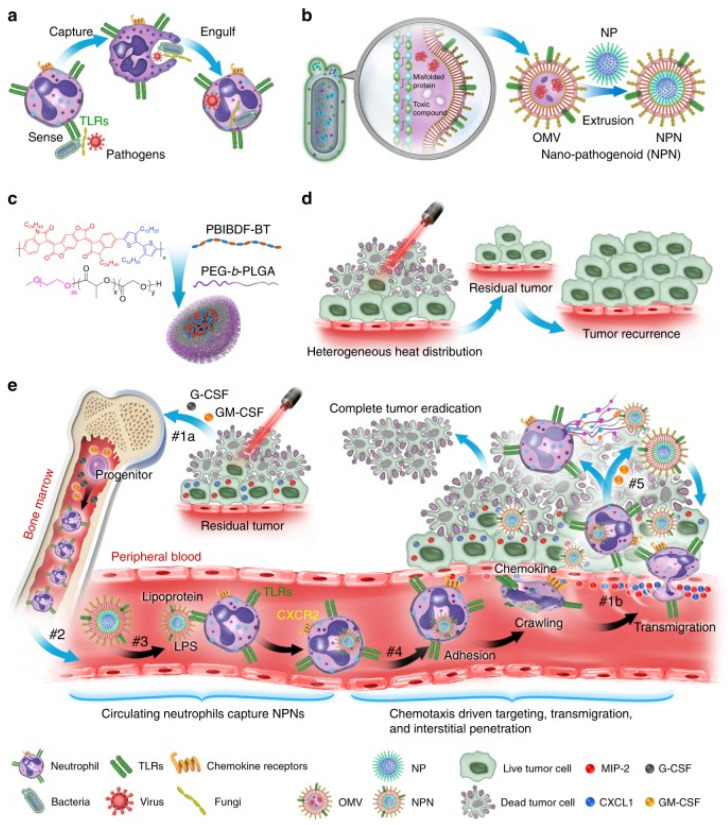
Schematic illustration showing the chemotaxis-driven delivery of NPNs for complete eradication of tumors post-phototherapy. (**a**) Neutrophils sense, capture, and engulf pathogens by recognizing the PAMPs with toll-like receptors (TLRs). (**b**) Preparation of NPNs by coating the OMVs on NPs, which inherit PAMPs from the OMVs. (**c**) Preparation of PEG-b-PLGA NPs encapsulating PBIBDF-BT (PBT) as a photothermal transducer. (**d**) The limited penetration of laser light used in PTT causes heterogeneous heat distribution within the tumor tissue and incomplete eradication of tumors, thus leading to tumor recurrence. (**e**) Treatment-induced cell death creates an inflammatory environment of the residual tumor and induces the production of granulocyte colony-stimulating factor (G-CSF), granulocyte–macrophage colony-stimulating factor (GM-CSF), and chemokines CXCL1 and MIP-2. #1a The released G-CSF and GM-CSF increase neutrophil production from bone marrow. #1b The released CXCL1 and MIP-2 broadcast the location of the inflamed tumor. #2 Neutrophils enter the blood circulation and encounter the injected NPNs. #3 Neutrophils sense NPNs with the recognition of LPS and lipoprotein by TLRs and subsequently engulf them. #4 Neutrophils laden with NPNs are recruited into the tumor site in response to the chemokine gradient through the following cascade: adhesion, crawling, and transmigration. #5 NPNs are released from neutrophils to kill tumor cells along with the formation of NETs in the inflamed tumor (Adapted from reference [36] with permission).

**Figure 2 cancers-14-04945-f002:**
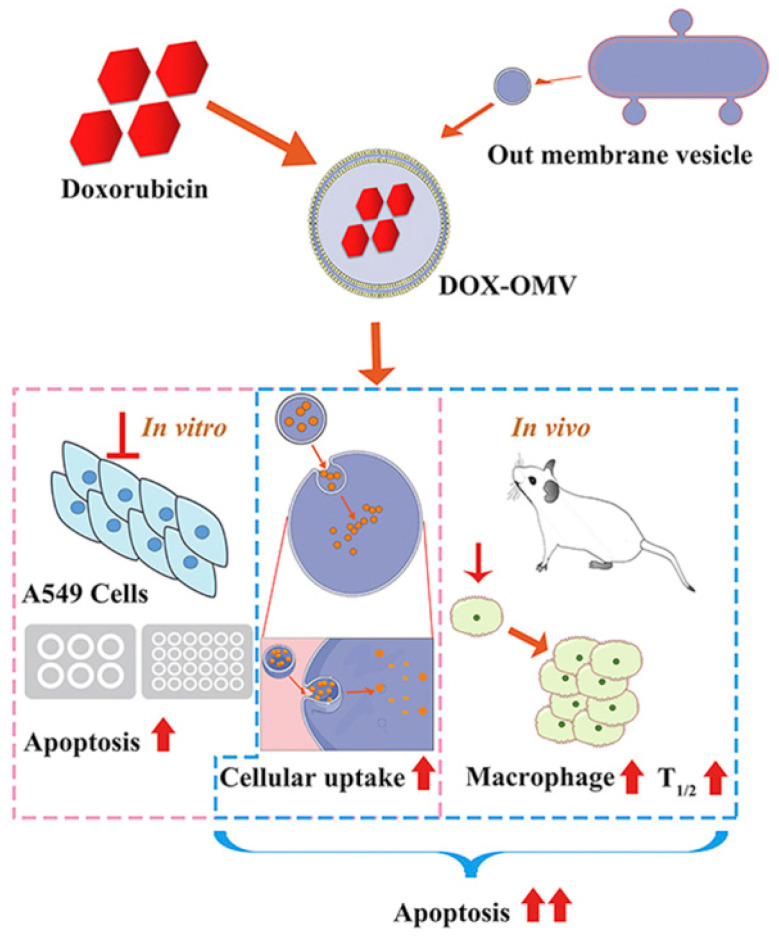
The attenuated *Klebsiella pneumonia* derived outer-membrane vesicles (OMVs), as a kind of biological drug-delivery carriers, are highly effective in transporting the chemotherapy drug doxorubicin (DOX) into nonsmall-cell lung cancer (NSCLC) A549 cells. Moreover, they can elicit appropriate immune responses, thereby enhancing the anti-NSCLC effect of DOX with no obvious toxicity in vivo (Adapted from reference [42] with permission).

**Figure 3 cancers-14-04945-f003:**
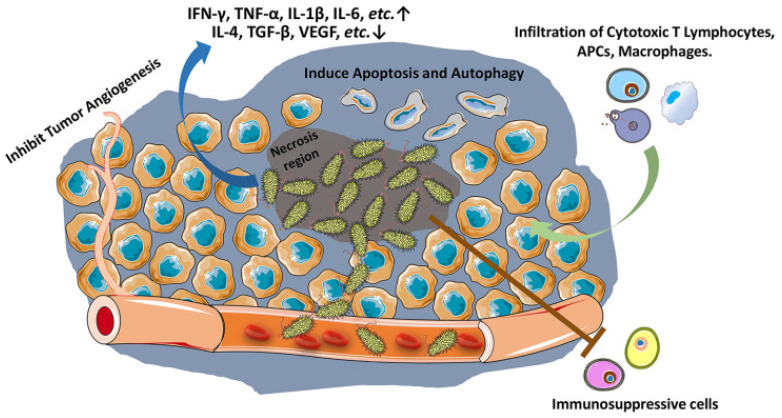
*Salmonella* stimulates host immune response against tumors. *Salmonella* accumulates in tumors (especially in necrosis region), inhibits tumor angiogenesis, and induces apoptosis and autophagy in tumor cells. *Salmonella* increases and activates cytotoxic T lymphocytes, antigen presenting cells (APCs) and macrophages against tumor cells, reduces tumor infiltration of Treg cells, and ablates the immunosuppressive capacity of myeloid-derived suppressor cells (MDSCs) and tumor-associated macrophages (Adapted from reference [61] with permission).

**Figure 4 cancers-14-04945-f004:**
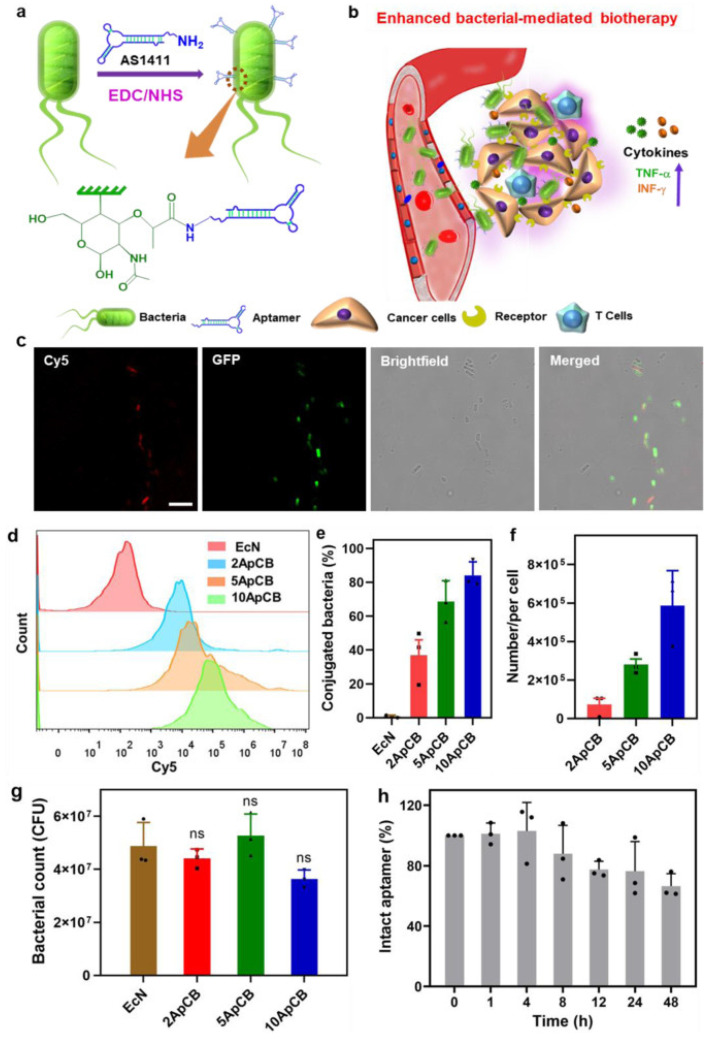
Design, preparation and characterization of ApCB. (**a**) Preparation of ApCB through amide condensation. (**b**) Aptamer-assisted tumor localization of bacteria for enhanced biotherapy. (**c**) Typical LSCM images of aptamer-conjugated bacteria. The red and green channels indicate aptamers conjugated with Cy5- and EcN-producing GFP, respectively. Images are representative of three independent biological samples. Scale bar: 10 μm. (**d**) Flow cytometric analysis of EcN and EcN conjugated with Cy5-labeled AS1411. (**e**) Percentages of conjugated EcN under different feed ratios. Error bars represent the standard deviation (n  =  3 independent experiments). Data are presented as mean values ± SD. (**f**) Average binding number of aptamers on each bacterial quantified by calculating the difference of fluorescent intensity of the aptamer solution after reaction. Error bars represent the standard deviation (n  =  3 independent experiments). Data are presented as mean values ± SD. (**g**) Bacterial viabilities of EcN, 2ApCB, 5ApCB, and 10ApCB by LB agar plate counting. Plates were incubated at 37 °C for 24 h prior to enumeration (n = 3 independent experiments). Data are presented as mean values ± SD; significance was assessed using Student’s t test (two-tailed); ns: no significance. (**h**) Degradation kinetics of the conjugated AS1411 in 90% phosphate-buffered serum solution at 37 °C. Error bars represent the standard deviation (n  = 3 independent experiments). Original data are provided as a Source Data file. (Adapted from reference [115] with permission).

**Figure 5 cancers-14-04945-f005:**
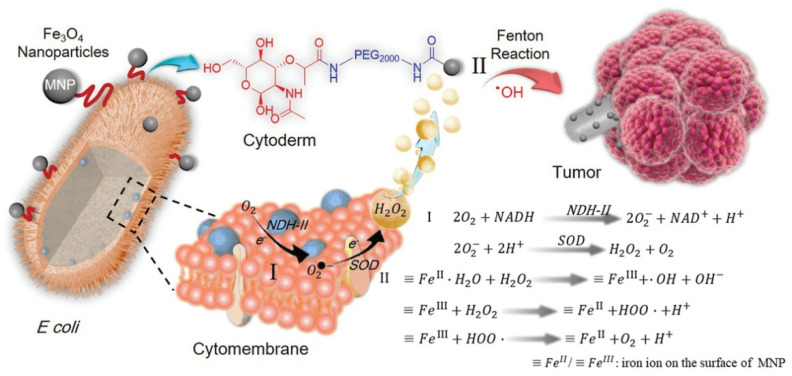
The scheme of bacteria-based Fenton-like bioreactor and its chemodynamic therapy process for antitumor therapy (Adapted from reference [116] with permission).

**Figure 6 cancers-14-04945-f006:**
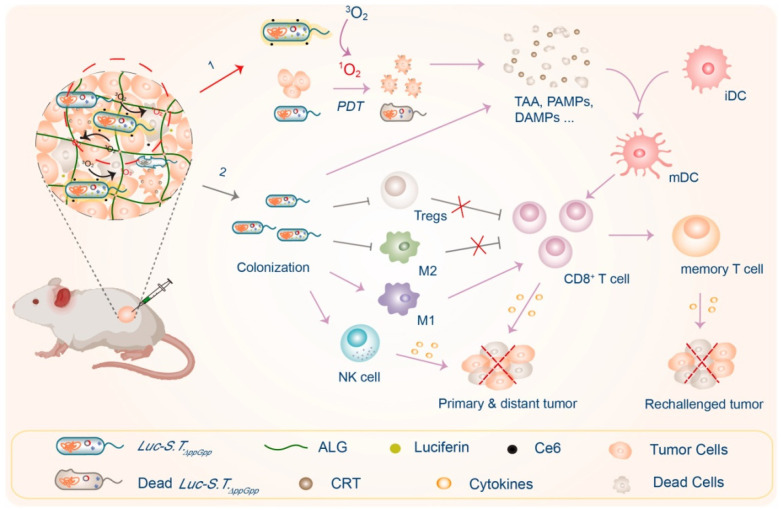
A scheme illustrating the engineering of bioluminescent bacteria to boost PDT and antitumor immunity for synergistic cancer treatment. Upon i.t. injection, engineered Luc-S.T.ΔppGpp would rapidly colonize and emit bioluminescence in the presence of substrate D-luciferin as the light source to boost PDT by activating Ce6, thereby causing cell death of both cancer cells and Luc-S.T.ΔppGpp itself to release tumor associate antigens (TAAs), DAMPs (e.g., CRT), and PAMPs (1). Meanwhile, such Luc-S.T.ΔppGpp colonization could also efficiently reverse the immunosuppressive tumor microenvironments (TMEs) by promoting intratumoral frequencies of M1 macrophages and NK cells, while suppressing intratumoral frequencies of M2 macrophages and Tregs (2). As the result, such Luc-S.T.ΔppGpp as both implantable light source (in the presence of D-luciferin) and immunostimulator could elicit potent innate and adaptive antitumor immunity to effectively suppress the growth of treated tumors, inhibit tumor metastasis, and prevent against tumor recurrence (Adapted from reference [130] with permission).

**Figure 7 cancers-14-04945-f007:**
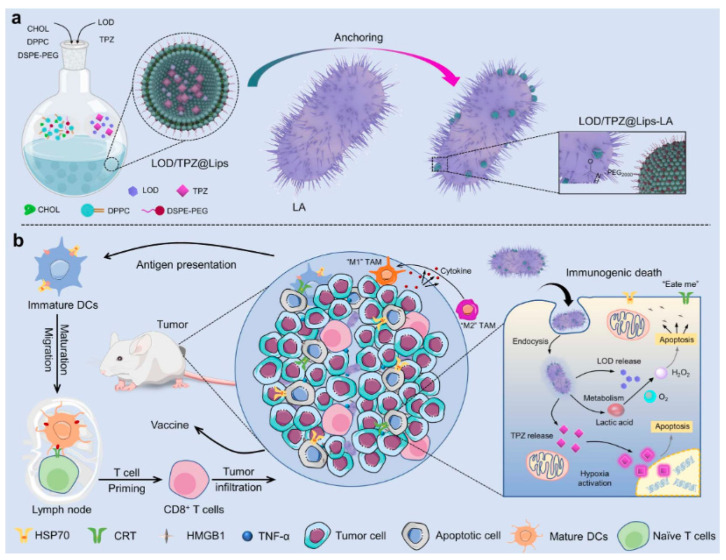
Schematic illustration of (**a**) the construction of LOD/TPZ@Lips-LA microbiotic nanomedicine by bonding LA and LOD co-loaded liposome onto the lactobacillus (LA) and (**b**) LOD/TPZ@Lips-LA triggered immunogenic cell death (ICD) and immune activation in tumor in synergy with the TZP-triggered chemotherapy (Adapted from reference [135] with permission).

**Figure 8 cancers-14-04945-f008:**
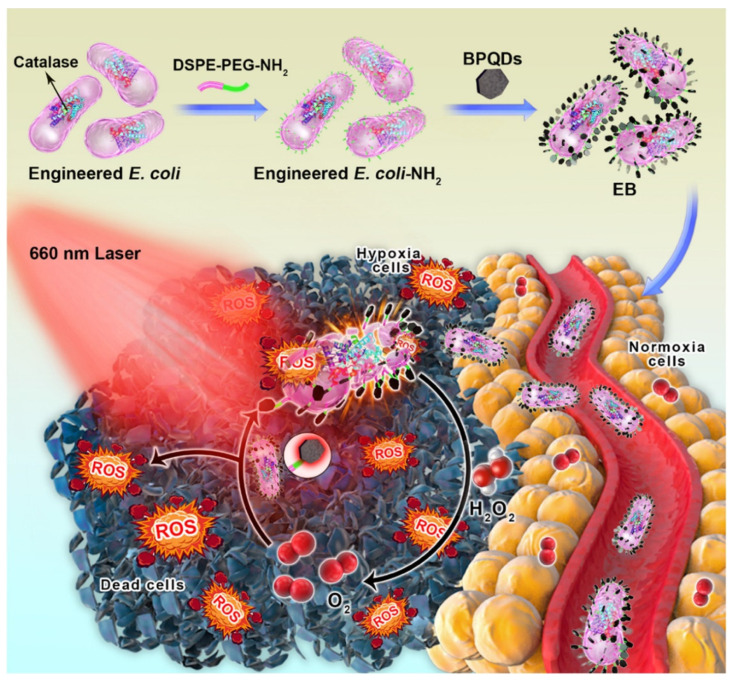
Schematic illustration of a novel engineered bacterium/black phosphorus quantum dot hybrid system for hypoxic tumor targeting and efficient photodynamic therapy (Adapted from reference [137] with permission).

**Figure 9 cancers-14-04945-f009:**
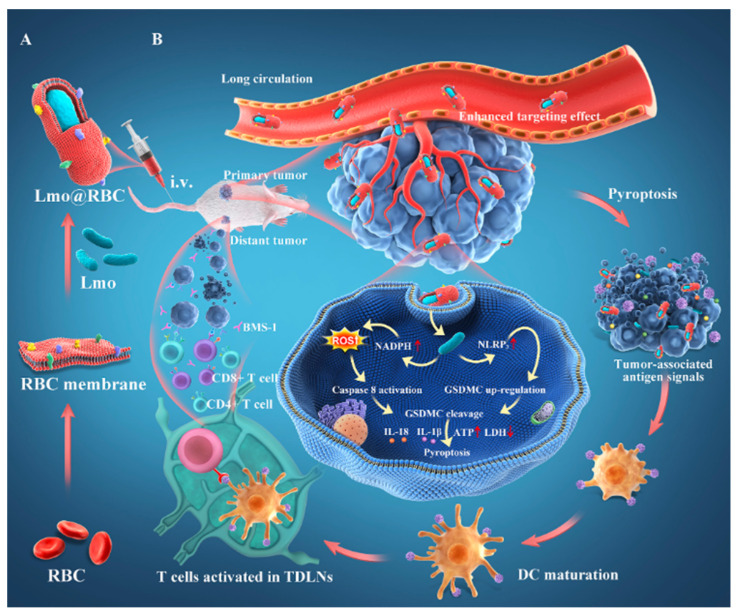
Schematic depiction of utilizing LMO@RBC to improve cancer immunotherapy. (**A**) Schematic illustration of RBC membranes extraction and the preparation of LMO@RBC. (**B**) Tumor-homing LMO@RBC effectively accumulated in primary tumor after intravenous administration and triggers cancer cell pyroptosis. Pyroptotic cancer cells in primary tumor release proinflammatory substances, which induce DC maturation and T cell activation in tumor-draining lymph nodes (TDLNs), resulting in efficient suppression of primary and remote tumors (Adapted from reference [138] with permission).

**Table 1 cancers-14-04945-t001:** Biosurfactants with antitumor activity against cancer cells.

Biosurfactant		Cancer Type	References
Cyclic lipopeptide	*Bacillus subtilis natto TK-1*	Breast cancer	[22]
Surfactin	*Bacillus subtilis natto T-2*	Breast cancer	[23]
L-lysine biopolymer Epsilon-poly-L-lysine	*Marine Bacillus subtilis* sp.	Liver carcinoma Cervix adenocarcinoma	[24]
viscosin	*Pseudomonas libanensis m9-3*	Breast cancer	[25]
AT514	*Serratia marcescens*	B-cell chronic lymphocytic leukemia	[27]
BE18591	*Streptomyces* sp.	Gastric cancer	[28]
Roseophilin	*Streptomyces* sp.	Hematologic cancer Colon cancer	[29]

**Table 2 cancers-14-04945-t002:** Ongoing and previous clinical trial details on bacterial strain alone or in combination for cancer treatment (Adapted from reference [71] with permission).

Bacterial Strain	Phase	Cancer Type	Number of Patients	References
*Salmonella typhimurium* VNP20009 (attenuated Salmonella typhimurium)	I	Metastatic melanoma; metastatic renal cell carcinoma	25	[72]
*S. typhimurium* VNP20009 (live genetically modified *S. typhimurium*)	I	Melanoma	4	[73]
*S. typhimurium* VNP20009 (attenuated Salmonella bacterium expressing the E. coli cytosine deaminase gene)	I	Head and neck or esophageal adenocarcinoma	3	[74]
*S. typhimurium* VNP20009 (live, genetically modified Salmonella typhimurium	I	Patients with advanced or metastatic solid tumors	Not provided	NCT00004216 [75]
*S. typhimurium* VNP20009 (live, genetically modified Salmonella typhimurium)	I	Unspecified adult solid tumors	Not provided	NCT00006254 [76]

*S. typhimurium* VNP20009 (live, genetically modified Salmonella typhimurium)	I	Neoplasm or neoplasm metastatic tumors	45	NCT00004988 [77]

*S. typhimurium* (IL-2 expressing, attenuated S. typhimurium)	I	Liver cancer	22	NCT01099631 [78]
*S. typhimurium* Ty21a VXM01 (live attenuated S. typhi carrying an expression plasmid encoding VEGFR)	I	Pancreatic cancer	26	[79]
*Clostridium Novyi-NT* spores	I	Colorectal cancer	2	NCT00358397 [80]
*Clostridium Novyi-NT* spores	I	Solid tumor malignancies	5	NCT01118819 [81]
*Clostridium novyi-NT*	I	Solid tumor malignancies	24	NCT01924689 [82]
*C. novyi-NT* spores	Ib	Refractory advanced solid tumors	18	NCT03435952 [83]
*Listeria monocytogenes*	II	Metastatic pancreatic tumors	90	[84]
*L. monocytogenes*	II	Cervical cancer	109	[85]
*L. monocytogenes*	III	Cervical cancer	450-	NCT02853604 [86]

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
