# Peer review of "Recent Advances in Bacteria-Based Cancer Treatment"

_cancers, 2022, doi:10.3390/cancers14194945_

Round 1

Reviewer 1 Report

The review has a complete list of new bacteria-based cancer treatments, but I suggest increasing the information provided with the pros and cons of those therapies. The addition of case studies for some of the treatments may have a beneficial understanding of some of the treatments and show some ongoing data of those studies. 

Also, a table with the current trials will be very illustrative for this review. 

Author Response

Dear Reviewer,

Thank you for your valuable review and suggestions about our manuscript.

1)The first advice is that "increasing the information provided with the pros and cons of those therapies". We have undertaken the following modification:

Despite the advantages of bacteria-based anti-tumor therapy, it is still limited by a few shortcomings that need to be urgently settled down. The most important is effectiveness. Bacteria in anti-cancer treatment refer to precise targeting processes and complex human immune responses in tumor environments. Engineered bacteria need to cross complex blood vessels in a high-speed blood flow to reach the tumor environment after intravenous injection. How to accumulate enough engineered bacteria to exert anti-cancer effects is an urgent consideration. Ensuring the highly efficient expression of proteins that exert anti-cancer effects for a long time has an important impact on bacteria-based anti-cancer therapy. Another issue to be aware of is safety. Bacteria are known as an autonomously disorganized and proliferating species. Its unique PAMPs and virulence factors will induce an immune response in the human body, which promotes the killing of cancer cells by human immune system. When the immune response is excessive, it threatens the patient's life. Genetic material based on engineered bacteria also refers to ethical issues with transgenes, which presses challenges for the future of engineered bacteria therapy. 

2)The advice is that "The addition of case studies for some of the treatments may have a beneficial understanding of some of the treatments and show some ongoing data of those studies". We have undertaken the following modification:

The S. typhimurium VNP20009 strain and Clostridium novyi-NT spores have entered Phase I clinical trial, which all revealed the promising anti- tumor effect. The famous S. typhimurium VNP20009 strain achieved the purpose of constructing attenuated and purine-deficient strains through msbB and purL deletion. After intravenous injected 1 × 106–1 × 109 CFU/ml of S. typhimurium VNP20009 in 24 patients with metastatic melanoma, bacteria target purine-rich tumor regions and reduces the host's nitric oxide and proinflammatory cytokines such as TNF-α and IL-1β. Unfortunately, no objective anti-tumor effect was observed. Engineering bacteria to specifically target tumors or the combinations of bacteria-based with anti-tumor protein will be applicated in therapeutic effect on tumor in the future.

2)The advice is that "a table with the current trials". We have undertaken the following modification:

Table 2. Ongoing and Previous clinical trial details on bacterial strain alone or in combination for cancer treatment.

(Adapted from reference [71] with permission.)

Bacterial strain

Phase

Cancer type

Number of patients

References

Salmonella typhimurium VNP20009(attenuated Salmonella typhimurium)

I

Metastatic melanoma; metastatic renal cell carcinoma

25

[72]

S. typhimurium VNP20009(Live genetically modified S. typhimurium (VNP 20009).

I

Melanoma

4

[73]

S. typhimurium VNP20009 (attenuated Salmonella bacterium expressing the E. coli cytosine deaminase gene)

I

Head and neck or esophageal adenocarcinoma

3

[74]

S. typhimurium VNP20009 (Live, Genetically Modified Salmonella Typhimurium (VNP20009)

I

Patients with advanced or metastatic solid tumours

Not provided

NCT00004216

[75]

S. typhimurium VNP20009(Live, Genetically Modified Salmonella Typhimurium)

I

Unspecified adult solid tumours

Not provided

NCT00006254

[76]

-

S. typhimurium VNP20009(Live, Genetically Modified Salmonella Typhimurium)

I

Neoplasm or neoplasm metastatic tumors

45

NCT00004988

[77]

-

S. typhimurium (IL-2 Expressing, Attenuated S. typhimurium)

I

Liver cancer

22

NCT01099631

[78]

S. typhimurium Ty21a VXM01(live attenuated S. typhi carrying an expression plasmid encoding VEGFR)

I

Pancreatic cancer

26

            [79]

Clostridium Novyi-NT Spores

I

Colorectal cancer

2

NCT00358397

           [80]

Clostridium Novyi-NT Spores

I

Solid tumour malignancies

5

NCT01118819

[81]

Clostridium novyi-NT

I

Solid tumour malignancies

24

NCT01924689

[82]

C. novyi-NT spores

Ib

Refractory advanced solid tumours

18

NCT03435952

[83]

Listeria monocytogenes

II

Metastatic pancreatic tumours

90

[84]

L. monocytogenes

II

Cervical cancer

109

[85]

L. monocytogenes

III

Cervical cancer

450-

NCT02853604

[86]

Thank you for your positive view on our manuscript and your very constructive suggestions again. 

Best regards.

Reviewer 2 Report

In the manuscript entitled “Recent advances in bacteria-based cancer treatment” the authors bring the state of art of bacteria-based cancer treatment bringing to light the advantages and the risks that must be overcome. Also pointed out the techniques used to reduce potential bacterial virulence, what could allow this approach To be used.

The subject is interesting and relevant, and the manuscript is mostly well written; however, the manuscript possesses some writing issues that need to the improved:

1-      In the section 2.3 biosurfactant, all that descriptions could be summarized in a table, while the text could concentrate in better describing the cycle Lipopeptide and its role in Bacillus subtilis natto TK1 and pointing the main aspects of its importance regarding cancer tratment. Even though it´s a matter of writing style, this Reviewer believe that this would make the information clearer since I found myself curious and felt the lack of this information at the beginning of the paragraph. This was correctly done by the authors on the begining of the section 2.4. Extracellular surface.

2-      The first paragraph of the section 3 has no references. Please include a proper reference.

3-      The section 3.4 Lacks references in the paragraph starting at line 322.

4-      Figure 4 has low resolution.

 Thus, this reviewer does RECOMMEND this paper for publication if the authors do consider these changes.

Author Response

Dear Reviewer,

Thank you for your valuable review and suggestions about our manuscript.

1)The first advice is that "In the section 2.3 biosurfactant, all that descriptions could be summarized in a table". We have undertaken the following modification:

The Table 1 demonstrated biosurfactants with cancer cell proliferation, which is known as antitumor agents and inhibits some cancer progression processes. 

Table 1. Biosurfactants with antitumor activity against cancer cells

Biosurfactant

Cancer type

References

Cyclic lipopeptide

Bacillus subtilis natto TK-1

Breast cancer

[22]

Surfactin

Bacillus subtilis natto T-2

Breast cancer

[23]

L-lysine biopolymer Epsilon-poly-L-lysine

Marine Bacillus subtilis sp

 Liver carcinoma

Cervix adenocarcinoma

[24]

viscosin

Pseudomonas libanensis m9-3

Breast cancer

[25]

AT514

Serratia marcescens

B-cell chronic lymphocytic leukemia

[27]

BE18591

Streptomyces sp

Gastric cancer

[28]

Roseophilin

Streptomyces sp

Hematologic cancer

Colon cancer

[29]

2)The advice is about "the text could concentrate in better describing the cycle Lipopeptide and its role in Bacillus subtilis natto TK1". We have undertaken the following modification: 

Xiaohong Cao et al. demonstrated that cyclic lipopeptide inhibited proliferation of human breast cancer MCF-7 cells by inducing apoptosis and increasing ion calcium concentration in the cytoplasm. Flow cytometric analysis revealed that cyclic lipopeptide caused dose- and time-dependent apoptosis through cell arrest at G(2)/M phase [22]. Another lipopeptide such as surfactin, have also been demonstrated their potential antitumor activity against several cancer cell lines. [23].

3)The advice is about "pointing the main aspects of its importance regarding cancer tratment". We have undertaken the following modification: 

Biosurfactant shows the promising application in microemulsion-based drug formulations. Microemulsion comprises an aqueous phase, an oil phase and a surfactant, which can encapsulate or solubilize a hydrophobic or hydrophilic drug for anti-tumor therapy. The combination of biosurfactant and liposome also demonstrates specifically targeted. Shim, Ga Yong et al. revealed that surfactin enhanced cellular delivery of liposome siRNA in Hela cells. In this way, it was possible to improve the antitumor effectiveness of those nanoparticles [31]. Biosurfactants have applicated in broad-spectrum anti-tumor treatments and are viewed as safe vehicles or ingredients in drug delivery systems.

4)The question is about "The first paragraph of the section 3 has no references.". We have undertaken the following modification: 

Compared with the extracted bacterial components, anaerobic or facultative anaerobic naïve living bacteria have better tumor targeted. When naïve living bacteria enter the host body, they can actively target the tumor microenvironment, which includes the characteristics of hypoxia, high purine and low acid content. They can deliver the anticancer bacterial components more efficiently. The bacteria with anticancer activity that have been verified by animal experiments include probiotics, Lactobacillus[51-58] and Escherichia coli[59], Streptococcus, Lactococcus lactis, Lactobacillus casei, Lactobacillus casei Zhang, Bifidobacterium longum, Clostridium butyricum, etc. General toxic bacteria include Mycobacterium tuberculosis[60], Salmonella typhimurium[61], Listeria monocytogenes[62] and pathogenic bacteria such as Vibrio parahaemolyticus, Pseudomonas aeruginosa[63], etc.

5)The question is about "The section 3.4 Lacks references in the paragraph starting at line 322." We have undertaken the following modification:

  1. Zheng, D.W., et al., Prebiotics-Encapsulated Probiotic Spores Regulate Gut Microbiota and Suppress Colon Cancer. Adv Mater, 2020. 32(45): p. e2004529.

 5)The question is about "Figure 4 has low resolution." We have replaced the figure with a higher resolution.

Thank you for your positive view on our manuscript and your very constructive suggestions again. 

Best regards.

Reviewer 3 Report

In this review, the authors discussed the anticancer activity of bacterial cellular components, the direct anticancer effect of naïve living bacteria, the anti-tumor mechanism, and the latest research progress in detail. Then, they summarize the latest research progress of engineering bacteria in recent years in order to solve the limitations of bacteria-based anti-tumor treatment. At last, the authors point out that the future development directions of bacterial tumor therapy are to combine nanomaterials and engineering bacteria to acquire a more specific and effective immune response with less toxic side-effect to the living system. The content of the paper is extensive and clear, and each viewpoint is elaborated in combination with basic research and clinical trials, pointing out the difficulties and challenges faced by future research for bacteria-based tumor therapy. I support the publication of this review in cancers. There are also some minor issues.

1.       In terms of discussion and outlook, it is suggested to discuss the link between bacteria and clinical applications. It is suggested to specifically discuss the prospects of bacteria in cancer treatment.

2.       References updated to the last five years whenever possible.

3.       What is the anti-tumor effect of the normal flora of the human body?

4.       Chapter 4.5 is simply a list of literature without own opinions. For example, what are the advantages and shortcomings for engineered bacteria?

5.       Due to the large content of this review, it is recommended that the author give some highlights, such as “possible future perspectives in bacteria-based cancer immunotherapy” or what the author thinks is more important.

Author Response

Dear Reviewer,

Thank you for your valuable review and suggestions about our manuscript.

1)The first advice is that " In terms of discussion and outlook, it is suggested to discuss the link between bacteria and clinical applications. It is suggested to specifically discuss the prospects of bacteria in cancer treatment". We have undertaken the following addition: 

The clinical trials for bacterial cancer treatments are summarized in Table 2. The S. typhimurium VNP20009 strain and Clostridium novyi-NT spores have entered Phase I clinical trial, which all revealed the promising anti- tumor effect. The famous S. typhimurium VNP20009 strain achieved the purpose of constructing attenuated and purine-deficient strains through msbB and purL deletion. After intravenous injected 1 × 106–1 × 109 CFU/ml of S. typhimurium VNP20009 in 24 patients with metastatic melanoma, bacteria target purine-rich tumor regions and reduces the host's nitric oxide and proinflammatory cytokines such as TNF-α and IL-1β. Unfortunately, no objective anti-tumor effect was observed. Engineering bacteria to specifically target tumors or the combinations of bacteria-based with anti-tumor protein will be applicated in therapeutic effect on tumor in the future.

2)The second advice is that "References updated to the last five years whenever possible". We have added references and now there are 60 references published in the last five years.

3)The question is that "What is the anti-tumor effect of the normal flora of the human body". We have explained in chapter 2 and chapter 3. Detail as following:

2 Bacterial components of anti-tumor treatment

To date, bacterial toxins produced by bacterial cells, such as the Coley toxin, diphtheria toxin, Clostridium perfringens enterotoxin, bacterial enzymes L-asparaginase and arginine deaminase, and biosurfactant, such as surface and prodigiosin-like pixels, is able to effectively inhibit tumor growth through cell cycle arrest, tumor cell signal pathway interruption and other mechanisms. In addition, the components of bacteria, bacterial outer surface, the bacterial membrane, bacterial wall, and biofilm can also specifically activate the immune response to kill tumor cells.......

3.4 lactic acid bacteria

Lactic acid bacteria are a kind of probiotic Gram-positive bacteria that include Lactococcus and Lactobacillus. Spherical lactic acid bacteria include Streptococcus, Lactococcus lactis, and Pediococcus. Lactobacilli include Lactobacillus rhamnosus, Lactobacillus casei, and Bifidobacterium longum. Streptococcus is a probiotic that kills tumor cells by activating host immunity and has been verified in animal experiments and clinical experiments [51]. Lactococcus lactis produces Nisin A, inhibits the growth of colon cancer tumors, and stops the cell division cycle [52]. Lactobacillus casei produces antimicrobial peptide kl15; interleukin (IL) -22 cytokines are downregulated, and caspase-7 and caspase-9 are upregulated, inducing apoptosis of colon cancer cells and the host immune response [52]. The oral probiotics Lactobacillus casei Zhang increase the levels of short chain fatty acids and nicotinamide in the serum and kidney, which reduces the damage to kidney cells [53]. Lactococcus lactis and Streptococcus bovis produce Nisin A and bovicin HC5, respectively, which kill breast cancer cells. In addition, antimutagenic and anti-inflammatory effects from lactic acid bacteria were also found [54, 55].

Bifidobacterium longum is a facultative anaerobic probiotic strain. After intravenous injection of mice, it was found to specifically locate solid tumors and slow down tumor growth. Anticancer treatment seems to increase TNF-α Cytokine and nitric oxide synthesis [56-58].

Spores-dex are prepared by the chemical reaction between Clostridium butyricum and glucan. The spores-dex can specifically target colon cancer after oral administration. In the tumor microenvironment, Clostridium butyricum ferments glucan to produce anticancer short chain fatty acids. In the subcutaneous tumor model of mice, the high-efficiency tumor inhibition ability of drug-loaded spores-dex was verified [68].

4)The advice is that "Chapter 4.5 is simply a list of literature without own opinions. For example, what are the advantages and shortcomings for engineered bacteria". We have undertaken the following modification:

chapter 4: Engineered bacteria refer to expression exogenous protein in precise period and position. Despite the advantages of this, it is still limited by a few shortcomings that need to be urgently settled down. The most important issue to be aware of is safety. Engineered bacterial therapeutics based on engineered bacteria refers to obtention of antibiotic resistance cassettes and ethical issues with transgenes, which presses challenges for the future of engineered bacteria therapy. Although direct injection of bacteria has an obvious killing effect on cancer cells, its side effects cannot be ignored. The research has shown that too high or too low a number of bacteria will cause bacterial ecological imbalance [69], and the toxicity of bacteria is also harmful to normal tissues. The study from Cayetano Pleguezuelos-Manzano found that the occurrence of CRC in colorectal cancer was related to colistin produced by pathogenic PKS+ E. coli. It is a toxin encoded and synthesized by a PKS island, which induces DNA double bond breakage and the death of host cells [70]. Bacteria are known as an autonomously disorganized and proliferating species. Its unique PAMPs and virulence factors will induce an immune response in the human body, which promotes the killing of cancer cells by human immune system. When the immune response is excessive, it threatens the patient's life. In short, safe, and reliable attenuated targeted bacteria, gene editing to express endogenous bacterial toxins, pigment proteases, etc., or exogenous drug precursor enzymes, antigen immune fragments, cytokines, anti-immune checkpoints, non-coding RNA, and so on are used. It can push the bacterial treatment of cancer to a new level of low dose and high efficiency. 

chapter 5: Bacteria and nanomaterials are directly anti-tumor through the covalent connection of chemical amide bonds and can also be adsorbed together through electrostatic interaction. In addition, bacterial membrane fragments and outer membrane vesicles can be fused with bacteria through repeated freezing and thawing, ultrasound, extrusion, and other methods. The new combination of nanomaterials and bacteria can inhibit the growth of tumors through the anticancer activities of nanomaterials such as photothermal and photodynamic therapy, which not only reduce the toxicity to normal cells, but also increase the tumor targeted.

5)The advice is that "Due to the large content of this review, it is recommended that the author give some highlights, such as “possible future perspectives in bacteria-based cancer immunotherapy” or what the author thinks is more important". We have undertaken the following modification:

Despite the advantages of bacteria-based anti-tumor therapy, it is still limited by a few shortcomings that need to be urgently settled down. The most important is effectiveness. Bacteria in anti-cancer treatment refer to precise targeting processes and complex human immune responses in tumor environments. Engineered bacteria need to cross complex blood vessels in a high-speed blood flow to reach the tumor environment after intravenous injection. How to accumulate enough engineered bacteria to exert anti-cancer effects is an urgent consideration. Ensuring the highly efficient expression of proteins that exert anti-cancer effects for a long time has an important impact on bacteria-based anti-cancer therapy. Another issue to be aware of is safety. Bacteria are known as an autonomously disorganized and proliferating species. Its unique PAMPs and virulence factors will induce an immune response in the human body, which promotes the killing of cancer cells by human immune system. When the immune response is excessive, it threatens the patient's life. Genetic material based on engineered bacteria also refers to ethical issues with transgenes, which presses challenges for the future of engineered bacteria therapy. In general, the future development directions of bacterial tumor therapy will be to combine nanomaterials and engineering bacteria and to explore the physical and functional relationship between bacteria and nanomaterials. The results will be a more specific, effective, and accurate tumor immune response and a comprehensive and less toxic treatment system.